# Co-Speech Gesture Generation via Audio and Text Feature Engineering

Geunmo Kim
Korea Electronics Technology
Institute
Republic of Korea
rootmo96@keti.re.kr

Jaewoong Yoo
Korea Electronics Technology
Institute
Republic of Korea
jaewoong.yoo@keti.re.kr

Hyedong Jung
Korea Electronics Technology
Institute
Republic of Korea
hudson@keti.re.kr

## ABSTRACT

In recent years, the field of human-computer interaction (HCI) research has seen increasing efforts to model social intelligence and behavior based on artificial intelligence. For human-agent communication to evolve in a human-way, non-verbal features can be used as important factors. We conducted our research as part of the GENEA Challenge 2023[13], where the task is to generate human gestures using these non-verbal elements. We applied two main approaches to generating natural gestures. First, we modified the provided baseline model to apply RoBERTa-based speech transcription embedding, and second, we designed a gesture generation model by adding a zero-crossing rate and rhythmical features to the input features. The gestures generated by this method were evaluated as unnatural in terms of human-like and conformity. However, through this, we will study the SOTA model structure of gesture generation in the future and apply various preprocessing methods to the input data to generate natural gestures.

## CCS CONCEPTS

• **Human-centered computing → Human computer interaction (HCI)**.

## KEYWORDS

Human-Computer Interaction (HCI), Gesture Generation, Deep Learning, Multimodal Learning

**ACM Reference Format:**
Geunmo Kim, Jaewoong Yoo, and Hyedong Jung. 2023. Co-Speech Gesture Generation via Audio and Text Feature Engineering. In *INTERNATIONAL CONFERENCE ON MULTIMODAL INTERACTION (ICMI '23 Companion), October 9–13, 2023, Paris, France.* ACM, New York, NY, USA, 6 pages. https://doi.org/10.1145/3610661.3616553

## 1 INTRODUCTION

In recent years, the field of Human-Computer Interaction (HCI) research has seen an increase in efforts to model social intelligence and behavior based on artificial intelligence[2, 3]. According to Albert Mehrabian's Three elements of communication[20], humans

rely more on para-verbal and non-verbal elements of communication than on verbal elements. In order for human-agent communication to evolve towards the human-way, para-verbal and non-verbal behavioral cues can be used as important elements. People usually express social signals and behaviors through non-verbal behavioral cues such as facial expressions, body postures and gestures, or para-verbal behavioral cues such as tone and pitch from vocal sounds[26]. According to Vinciarelli et al. (2009)[26], 90% of nonverbal behavioral cues are associated with speech. Therefore, assuming that a matching gesture exists based on audio and speech data, we will participate in the GENEA Challenge 2023 and proceed with the co-speech gesture generation task. The generated co-speech gestures can be utilized for multi-modal fusion by considering matching and combining verbal, para-verbal, and non-verbal features in future research on human-agent communication.

In traditional gesture generation research, motion system frameworks have been proposed as concatenative approaches such as motion graphs[10]. In recent years, learning-based approaches have been used to generate high-quality and interactive gestures by utilizing neural networks such as FFNNs, RNNs, GANs, and VAEs[6, 8, 11, 22, 24]. There are also studies on gesture generation tasks using text, speaker identity and style, and personality parameters as input features for generation models[1, 12, 23, 27]. In GENEA Challenge 2023, our team applied two main approaches to achieve a more natural and appropriate matching with speech. First, we modified the provided baseline model with RoBERTa-based embedding for speech transcription, and second, we designed a gesture generation model by adding a zero-crossing rate and rhythmical feature as additional audio features to the input features.

As a result, it was evaluated as unnatural for human-likeness and appropriateness. After checking with a 3D animation tool, we found that there were some natural gestures, but most of them were inappropriate for speech. Through this experiment, we realized that using more features does not always lead to better generation performance.

## 2 BACKGROUND AND PRIOR WORK

### 2.1 Data-driven gesture generation research

Data-driven gesture generation models are models that learn from a large amount of data, such as audio, text, and pose data, and generate gestures that correspond to the data. There are a variety of studies [7][18][19][29] that use data-driven generative models to generate gestures.

Habibie, Ikhsanul, et al [7] combined the benefits of database matching and adversarial learning to generate 3D gestures. The paper used the k-Nearest Neighbors (k-NN) algorithm to consider the

similarity between the correct audio-pose data stored in the database and the input data. Based on this, the correct audio-pose data stored in the database is sequentially searched to find the data with the highest similarity to the input data. Then, a Conditional Generative Adversarial Network (cGAN) model[21] was used to generate gestures corresponding to the input data. Unlike the GAN model, the cGAN model can use additional information such as the label of the input data to generate the desired data while the generator and discriminator are training. Therefore, the paper used the results of the k-NN algorithm as additional information to generate gestures corresponding to the input data.

Lu, Shuhong, et al [18] used the encoder structure of Liu, Xian, et al [17] to extract features from text and audio, and the Vector-Quantized Variational AutoEncoder (VQ-VAE) model[25] to extract gesture features. The VQ-VAE model is a model that applies vector quantization (VQ) to the VAE model. Vector quantization is a technique that uses an algorithm similar to K-means clustering to replace continuous probability values with discrete values. By doing so, we converted the latent values of the gesture data into low-dimensional vectors. As a result, we generate gestures similar to the input data by learning low-dimensional latent variables that better represent the features of the gesture data.

Lu, Shuhong, et al [19] considered the problem that when generating gestures based on speech data, multiple gestures may be generated for the same speech data. To solve this problem, they used individual gesture tokens and a Residual-Quantized Variational Audoencoder (RQ-VAE) model[14]. By using discrete gesture tokens, we solved the mapping problem of gesture generation by assigning different probabilities to different gestures generated based on the same speech data. We also used the RQ-VAE model to train the discrete gesture tokens. The RQ-VAE model recursively discretizes the latent variables in the input data to reduce the loss of information as the encoding progresses. This resulted in higher-quality gestures.

Zhang, Fan, et al [29] proposed the DiffMotion model based on the diffusion model for gesture generation. The DiffMotion model consists of an Autoregressive Temporal Encoder (AT-Encoder) and a Denoising Diffusion Probabilistic Module (DDPM). The AT-Encoder uses a multi-layer LSTM structure to encode the temporal context of the speech data. Then, through the diffusion and generation process of the DDPM model, it learned a one-to-many mapping of input data and gestures and generated new gestures.

## 2.2 Multimodal gesture generation research

Multimodal-based research utilizes various types of data through multiple modalities to overcome the limitations of using only a single type of data for learning. Feature vectors are extracted using a deep learning structure suitable for each modality, and multiple tasks are performed based on them. Multimodal-based gesture generation research uses audio, text, and pose data as input data for each modality to extract feature vectors and utilize them to generate gestures that correspond to the input data. Various studies use this multimodal structure to generate gestures.

Kim, Gwantae, et al [9] proposed a new framework, Multimodal Pretrained Encoder for Feature generation (MPE4G), to generate natural gestures using (speech, text, motion) as input data for multimodal structures. This framework solves the problem of inaccurate gesture generation when there is noise in the input data used for training. To achieve this, the proposed framework consists of three main steps. First, a frame-by-frame embedder and generator are trained with joint embedding loss and reconstruction loss. Second, a multimodal encoder is trained with a self-supervised learning approach. Third, the embedder, encoder, decoder, and generator are jointly trained using supervised learning. Based on these components, we not only achieved good performance in gesture generation but also solved problems such as noise in the input data and generated natural gestures that respond to the input data.

## 3 METHOD

Our model structure for gesture generation is based on [4]. Our model structure consists of an encoder, an attachment, and a decoder, as shown in the following figure 1.

The encoder consists of character embedding, three 1d convolution layers, and a bi-directional LSTM. When a one-hot vector is input, it is converted into an embedding vector through character embedding. It is then converted to an encoded feature through a convolutional layer and a bi-directional LSTM. Attention is the process of aligning what information to get from the encoder by using the encoded features from the encoder and the features generated at the previous point in the decoder's LSTM. In our model, we use a locality constraint attention like [4]. The decoder consists of two (Fully connected layer + ReLU), a uni-directional LSTM, a Fully connected layer, and five convolutional layers. The alignment feature information obtained through attention and the gesture feature generated at the previous time is used to generate the gesture feature at the next time. Through this process, gestures corresponding to the input data are generated.

For gesture generation, we built on the aforementioned model structure and focused on input features. First, to vary the text features, we used RoBERTa-based (784 dimensions) pretrained with word embeddings. Next, we used mfcc, mel-spectrogram, pitch, and energy, which are commonly used audio features, as well as zero-crossing rate and rhythmical features.

We used two NVIDIA A100-SXM4-80GB GPUs to train the aforementioned models. For both Monadic and Dyadic, we trained for a total of 25,000 iterations and set the learning rate to 1e-4. We also used a weight decay value of 1e-6 and a batch size of 64 to match the GPU memory. For the optimizer and loss function used for training, we used the most popular Adam optimizer and MSE loss function.

## 3.1 Data and data processing

We trained our model using a dataset [15] provided by GENEA Challenge 2023. The dataset is based on the Talking With Hands 16.2M gesture dataset, which are audio and motion capture data of several pairs of people talking freely about various topics. The dataset consists of 372 training datasets and 41 validation datasets. The training and validation datasets contain motion capture data (BVH format), audio (WAV format), and transcript (CSV format) data corresponding to the motion, and speaker id (CSV format) data, respectively. Since GENEA Challenge 2023[13] considers not

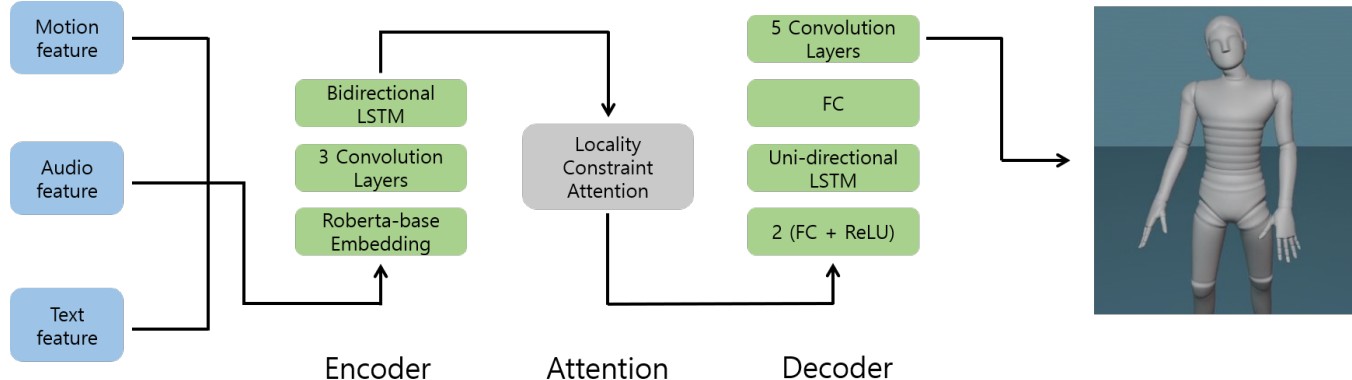

**Figure 1: Our Proposed Architecture**

only monadic but also dyadic situations, unlike GENEA Challenge 2022[28], the training and validation datasets include the main-agent and additionally the interlocutor.

*3.1.1 Motion.* We extracted features from the motion using PyMo library for gesture generation. The motion FPS is 30. The team used an exponential map[5] to represent 3D motion. Unlike GENEA Challenge 2022, GENEA Challenge 2023 evaluates only the full body[13]. Therefore, we utilised the motion features corresponding to the full body using the root position and 19 keypoints in the upper body and 6 keypoints in the lower body. Therefore, the full body has 78 dimensions.

*3.1.2 Audio.* We extracted several features from the audio for gesture generation. The sample rate of the audio is 44100 Hz. First, we used mfcc, mel-spectrogram, and prosody (energy, pitch) features, which are widely used in gesture generation research[16]. We also used zero-crossing rate and rhythmical feature in addition to the aforementioned features because we believe that gestures are highly related to audio. In the case of zero-crossing rate, the direction and shape of the gesture can be determined, so we thought that audio with a high zero-crossing rate could be used to generate gently waving gestures, etc. In the case of rhythmical feature, we thought that if the rhythm of the audio is uniform, the corresponding gesture will also have a smooth shape.

The characteristics of the six features mentioned above are as follows. For mfcc, mel-spectrogram, zero-crossing rate, and rhythmical features, the Librosa library was used. The prosody feature was extracted using the Parselmouth library. mfcc, mel-spectrogram, zero-crossing rate, and rhythmical features were all extracted using a hop length of 1470 on the audio. The mel-spectrogram was extracted by specifying the number of filter banks as 64, and the mfcc was extracted using 40 dimensions. Thus, the features extracted from the audio for model training are mfcc (40 dimensions), mel-spectrogram (64 dimensions), prosody (4 dimensions), zero-crossing rate (1 dimension), and rhythmical feature (384 dimensions).

*3.1.3 Text.* We used pretrained word embedding to extract features from the text for gesture generation. For word embedding, we used the RoBERTa-based model (784 dimensions). The RoBERTa-based model is a Transformer-based language model that performs better than BERT by applying several improvements. Unlike BERT, it does not use masking during the training process, which shortens the training time and improves performance. It also shows better generalization performance by using layer regularization, which is one of the techniques to prevent model overfitting during the training process. We used the RoBERTa-based model as our word embedding model.

The text features used to train the model were extracted using the transcripts contained in the provided dataset. Each text data was preprocessed with a word embedding model, and all OOV words were zeroed. In addition, we used metadata information such as the speaker's ID and the presence or absence of finger joints.

## 4 EVALUATION

GENEA Challenge 2023 was slightly different from GENEA Challenge 2022 in that it was evaluated on three different aspects:

- `Human-likeness`: How human-like the gestures are, regardless of the speech
- `Appropriateness for agent speech`: Evaluation of natural gestures for speech of the interlocutor, while considering human-likeness.
- `Appropriateness for the interlocutor`: Evaluate whether the interlocutor shows appropriate gestures to match the speech of the interlocutor, while considering human-likeness.

## 4.1 Result and Discussion

The test dataset used to compare and analyze the performance of our gesture generation model was provided by GENEA Challenge 2023. Unlike GENEA Challenge 2022, we also considered dyadic situations, so the dataset used to generate gestures for the main-agent includes motion, audio, and text data for the interlocutor. We submitted the motion data generated using the test dataset to GENEA Challenge 2023 for evaluation and received the following evaluation results.

*4.1.1 Human-likeness.* Table 1 shows the results of the human-likeness evaluation. Our submission falls into the SC submission,

and as can be seen in Table 1, it was evaluated as an unnatural gesture in terms of human-likeness. To analyze these results, we visualized some of the gestures generated by our model using a 3D animation tool called Blender. When we checked the visualized gestures, we found that our model produced several unnatural gestures, such as the gesture with the right arm fixed (left in Figure 2) and the gesture with the right arm bent behind the head (right in Figure 2), as shown in Figure 2. This confirmed that our model produced a large number of unnatural gestures, as shown in Table 1. We also confirmed that simply increasing the number of input features, which was the focus of our research, can have a detrimental effect on the model's ability to generate gestures by learning unnecessary information.

| Condi-tion | Human-likeness | |
|---|---|---|
| | Median | Mean |
| NA | $71 \in [70, 71]$ | 68.4±1.0 |
| SG | $69 \in [67, 70]$ | 65.6±1.4 |
| SF | $65 \in [64, 67]$ | 63.6±1.3 |
| SJ | $51 \in [50, 53]$ | 51.8±1.3 |
| SL | $51 \in [50, 51]$ | 50.6±1.3 |
| SE | $50 \in [49, 51]$ | 50.9±1.3 |
| SH | $46 \in [44, 49]$ | 45.1±1.5 |
| BD | $46 \in [43, 47]$ | 45.3±1.4 |
| SD | $45 \in [43, 47]$ | 44.7±1.3 |
| BM | $43 \in [42, 45]$ | 42.9±1.3 |
| SI | $40 \in [39, 43]$ | 41.4±1.4 |
| SK | $37 \in [35, 40]$ | 40.2±1.5 |
| SA | $30 \in [29, 31]$ | 32.0±1.3 |
| SB | $24 \in [23, 27]$ | 27.4±1.3 |
| SC | $9 \in [9, 9]$ | 11.6±0.9 |

Table 1: The table of statistics for the human-likeness evaluation, with confidence intervals at the level $\alpha = 0.05$. Conditions are ordered by decreasing sample median rating.

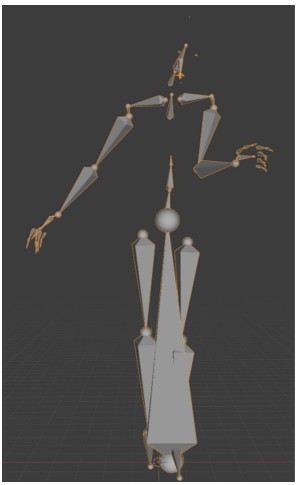 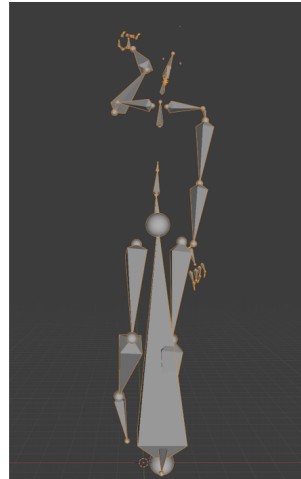

Figure 2: Visualisation of the unnatural generated gestures

*4.1.2 Appropriateness.* Table 2 shows the evaluation results in terms of appropriateness for speech. For our submission, SC, the evaluation result is an unnatural gesture that is not appropriate for speech in terms of appropriateness to speech. As with human-likeness, we visualized the generated gestures to analyze the evaluation results. When we checked the visualized gestures, we found that in many cases we were unable to generate gestures that corresponded to the speech. The evaluation results and visualizations confirmed that the zero-crossing rate and rhythmical features, which we used as additional input features, require different preprocessing.

| Condi-tion | 2*MAS | Pref. matched | Raw response count | | | | | |
|---|---|---|---|---|---|---|---|---|
| | | | 2 | 1 | 0 | −1 | −2 | Sum |
| NA | 0.81±0.06 | 73.6% | 755 | 452 | 185 | 217 | 157 | 1766 |
| SG | 0.39±0.07 | 61.8% | 531 | 486 | 201 | 330 | 259 | 1807 |
| SJ | 0.27±0.06 | 58.4% | 338 | 521 | 391 | 401 | 155 | 1806 |
| BM | 0.20±0.05 | 56.6% | 269 | 559 | 390 | 451 | 139 | 1808 |
| SF | 0.20±0.06 | 55.8% | 397 | 483 | 261 | 421 | 249 | 1811 |
| SK | 0.18±0.06 | 55.6% | 370 | 491 | 283 | 406 | 252 | 1802 |
| SI | 0.16±0.06 | 55.5% | 283 | 547 | 342 | 428 | 202 | 1802 |
| SE | 0.16±0.05 | 54.9% | 221 | 525 | 489 | 453 | 117 | 1805 |
| BD | 0.14±0.06 | 54.8% | 310 | 505 | 357 | 422 | 220 | 1814 |
| SD | 0.14±0.06 | 55.0% | 252 | 561 | 350 | 459 | 175 | 1797 |
| SB | 0.13±0.06 | 55.0% | 320 | 508 | 339 | 386 | 262 | 1815 |
| SA | 0.11±0.06 | 53.6% | 238 | 495 | 438 | 444 | 162 | 1777 |
| SH | 0.09±0.07 | 52.9% | 384 | 438 | 258 | 393 | 325 | 1798 |
| SL | 0.05±0.05 | 51.7% | 200 | 522 | 432 | 491 | 170 | 1815 |
| SC | −0.02±0.04 | 49.1% | 72 | 284 | 1057 | 314 | 76 | 1803 |

Table 2: The table of statistics for the speech appropriateness evaluation, with confidence intervals for the mean appropriateness score (MAS) at the level $\alpha = 0.05$. "Pref. matched" identifies how often test-takers preferred matched motion in terms of appropriateness, ignoring ties.

| Condi-tion | 2*MAS | Pref. matched | Raw response count | | | | | |
|---|---|---|---|---|---|---|---|---|
| | | | 2 | 1 | 0 | −1 | −2 | Sum |
| NA | 0.63±0.08 | 67.9% | 367 | 272 | 98 | 189 | 88 | 1014 |
| SA | 0.09±0.06 | 53.5% | 77 | 243 | 444 | 194 | 55 | 1013 |
| BD | 0.07±0.06 | 53.0% | 74 | 274 | 374 | 229 | 59 | 1010 |
| SB | 0.07±0.08 | 51.8% | 156 | 262 | 206 | 263 | 119 | 1006 |
| SL | 0.07±0.06 | 53.4% | 52 | 267 | 439 | 204 | 47 | 1009 |
| SE | 0.05±0.07 | 51.8% | 89 | 305 | 263 | 284 | 73 | 1014 |
| SF | 0.04±0.06 | 50.9% | 94 | 208 | 419 | 208 | 76 | 1005 |
| SI | 0.04±0.08 | 50.9% | 147 | 269 | 193 | 269 | 129 | 1007 |
| SD | 0.02±0.07 | 52.2% | 85 | 307 | 278 | 241 | 106 | 1017 |
| BM | −0.01±0.06 | 49.9% | 55 | 212 | 470 | 206 | 63 | 1006 |
| SJ | −0.03±0.05 | 49.1% | 31 | 157 | 617 | 168 | 39 | 1012 |
| SC | −0.03±0.05 | 49.1% | 34 | 183 | 541 | 190 | 45 | 993 |
| SK | −0.06±0.09 | 47.4% | 200 | 227 | 111 | 276 | 205 | 1019 |
| SG | −0.09±0.08 | 46.7% | 140 | 252 | 163 | 293 | 167 | 1015 |
| SH | −0.21±0.07 | 44.0% | 55 | 237 | 308 | 270 | 144 | 1014 |

Table 3: The table of statistics for the evaluation of appropriateness for the interlocutor, with confidence intervals for the mean appropriateness score (MAS) at the level $\alpha = 0.05$. "Pref. matched" identifies how often test-takers preferred matched motion in terms of appropriateness, ignoring ties.

Table 3 shows the results of our evaluation in terms of appropriateness, i.e., the ability to generate gestures that match the speech as information about the interlocutor is added. To analyze the evaluation results, we visualized the gestures generated by our model. We found that our model did not generate appropriate gestures for the interlocutor, but unnatural gestures that were not related to the interlocutor's information, such as monadic situations. We thought that this could be improved by resolving the aforementioned issues of human-likeness and appropriateness.

After analyzing the results of the previous evaluation, we found that gesture generation based on input features, which is the focus of our research, requires appropriate preprocessing for each feature rather than simply adding features. Although most of the evaluation results show unnatural gestures, we believe that our research has the potential for further development.

## 5 CONCLUSION AND FUTURE WORK

We conducted a study to generate gestures according to input data (motion, audio, text) based on the model structure of [4]. As mentioned earlier, we conducted experiments by changing the word embedding and adding audio features based on the existing model structure. We did not focus on improving the performance of the gesture generation model, but rather on checking how gestures are generated according to the input features. After training our model in this way, we found that it produced low-quality gestures when evaluated. Through these results, we confirmed that the preprocessing method for each feature is important, not just increasing the number of input features, and we have the following plans to improve the performance of gesture generation by conducting experiments with various research methods.

We will conduct experiments by changing SOTA models such as diffusion, RQ-VAE, and detailed hyper-parameters instead of simply using the model structure used in the past. We will also conduct experiments in a different way to compare and analyze the performance of gesture generation according to the input features we focused on. In the past, we simply added features to learn, but in the future, we will conduct experiments by segmenting the features of motion, audio, and text. For example, we will conduct experiments using only motion features, only audio features, and a combination of motion and audio features to see which features have the most impact on gesture generation.

## ACKNOWLEDGMENTS

This work was supported by Institute of Information communications Technology Planning Evaluation (IITP) grant funded by the Korea government(MSIT) (2022-0-00043,Adaptive Personality for Intelligent Agents)

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
