# OpenReview forum: "Co-Speech Gesture Generation via Audio and Text Feature Engineering"
_ACM.org/ICMI/2023/Workshop/GENEA_Challenge — GENEA Challenge 2023 Workshopproceeding_

### Official Review · Reviewer_WcDy · 2023-07-31
**The idea is good but further engineering effort is needed.**

**Rating:** 4
**Confidence:** 5

**Review:**

The authors proposes to use text embeddings of pretrained models, i.e., roBERTa, instead of traditional word vectors, i.e., FastText, to try to improve the naturalness and the matching between speech and the generated gestures. The authors also tries to include zero-crossing rate and what they called "rhythmical features" for the same goal.

Although similar approaches have been used in other previous researches, i.e., using pretrained models, a complete and comprehensive comparison study of the effects of adopting these pretrained models in gesture generation task, and a guidance on how should gesture generation incorporate pretrained models are missing. It is good to see that the authors are making efforts on this direction. Their results could be useful for building more powerful gesture generation systems.

Unfortunately, from the results it is obvious that the authors have failed in training their model to work. Their model's outputs are evaluated as very unnatural. It is hard to conduct ablation study when the model is not working normally. As a results, the paper may not provide valuable information for the relevant fields. Thus, I feel like this paper is below the acceptance threshold in its current form.

Lastly, the paper should provide descriptions for techniques whose reference is not provided, i.e., what is "rhythmical features"? Or just providing a reference will suffice.

---

### Official Review · Reviewer_SPwa · 2023-08-01
**Substantial improvements are needed in the revised version**

**Rating:** 5
**Confidence:** 5

**Review:**

[Paper Summary]

This paper introduces a framework receiving motion features, audio features, and text features for producing co-speech motions of a target agent. The framework comprises an encoder, an attention model, and a decoder. The model performance was verified with respect to human-likeness and appropriateness metrics. The experimental results indicated that the proposed approach yields lower accuracy compared to related works in the GENEA 2023 challenge. The authors commented that further efforts should be investigated to improve the performance of this approach.

[Comments to authors]

Although the proposed solution seems potential, I believe the paper needs substantial improvements. I include serval concerns that the author may consider in the revised version.
1. My primary concern is about the contributions and the novelties of the proposed approach. What are the differences between the proposed approach compared to the baseline model? The authors stated that “First, we modified the provided baseline model with RoBERTa-based embedding for speech transcription, and second, we designed a gesture generation model by 80 adding a zero-crossing rate and rhythmical feature as additional  audio features to the input features.”, but what is the motivation for such modifications? To improve the model performance? If that is the case, that should be clearly revealed by the experimental results.
2. The paper presentation is, overall, acceptable. However, the paper does not cover enough details that allow readers to understand the proposed approach. For instance:

   2.1. How did the author construct the locality constraint attention? Have any modifications been made compared to the baseline?

   2.2. Fig.1 does not give readers an overview of the proposed approach. The Talking with Hands 16.2M dataset contains social signals of both the target agent and the interlocutor, so which features that the authors were utilizing?

   2.3. Basic information such as loss function, training parameters should be included
3. The paper objective is not really clear. What findings and take-away messages can readers obtain from the paper?
For instance, on page 5, the authors stated, "Although most of the  evaluation results show unnatural gestures, we believe that our research has the potential for further development.” What is the "potential" that the authors mentioned?
Also, in conclusion, the authors claimed that “we did not focus on improving the performance of the gesture generation model, but rather on checking how gestures are generated according to the input features ” If so, did the author conduct an ablation experiment to examine the role of individual input features (motion, speech, and text transcription) in generated motions?

4. There are some typos in the paper. For instance, L189-192 on Page 2: “The alignment feature information obtained through attention and the gesture feature generated at the previous time is used to generate the gesture feature at the next time. ” did the author mean next motion frame instead of next time?

---

### Decision · Program_Chairs · 2023-08-04

**Decision:**

Accept (Workshop proceeding)

**Comment:**

Both reviewers have concerns on the novelty of the proposed system, and the paper also does not have enough details. On balance, the chairs think this paper is valuable for presenting and discussing at the workshop, and accept this paper to the Workshop ICMI track to be published in the Adjunct ICMI Proceedings. Please update the paper based on the reviews:
* Must cite the main challenge paper from organisers. We have given instructions on how to do this in the paper preparation guideline.
* Discuss the motivation of the proposed components.
* Try to add missing technical details.